# Neutrophil-to-Lymphocyte Ratio as a Predictor of Short-Term Functional Outcomes in Acute Ischemic Stroke Patients

**DOI:** 10.3390/ijerph20020898

**Published:** 2023-01-04

**Authors:** Min-Su Kim, Min Young Heo, Hyo Jin Joo, Ga Yang Shim, Jinmann Chon, Sung Joon Chung, Yunsoo Soh, Myung Chul Yoo

**Affiliations:** Department of Physical Medicine & Rehabilitation, Kyung Hee University Medical Center, College of Medicine, Kyung Hee University, Seoul 02447, Republic of Korea

**Keywords:** ischemic stroke, neutrophil-to-lymphocyte ratio, modified Barthel Index, MMSE, modified Rankin Scale, NIHSS, outcome assessment

## Abstract

Background: Neutrophil-to-lymphocyte ratio (NLR), a systemic inflammatory biomarker, has been associated with poorer outcomes in acute ischemic stroke patients. The present study was designed to expand these findings by investigating the association between NLR and short-term functional outcomes in acute ischemic stroke patients. Methods: This retrospective study evaluated patients within 7 days after the onset of acute ischemic stroke. Stroke severity on admission was measured using the National Institutes of Health Stroke Scale (NIHSS). The functional outcomes were assessed using the Berg Balance Scale (BBS), Manual Function Test (MFT), the Korean version of the modified Barthel Index (K-MBI), and the Korean Mini-Mental State Examination (K-MMSE) within 2 weeks of stroke onset. The modified Rankin Scale (mRS) was evaluated at discharge. Results: This study included 201 patients, who were grouped into three NLR tertiles (<1.84, 1.84–2.71, and >2.71) on admission. A multivariate analysis showed that the top tertile group (NLR > 2.71) had significantly higher risks of unfavorable outcomes on the K-MBI (*p* = 0.010) and K-MMSE (*p* = 0.029) than the bottom tertile group (NLR < 1.84). Based on the optimal cut-off values from a receiver operating characteristic curve analysis, a higher NLR was significantly associated with higher NIHSS scores (*p* = 0.011) and unfavorable outcomes on the K-MBI (*p* = 0.002) and K-MMSE (*p* = 0.001). Conclusions: A higher NLR is associated with poorer short-term functional outcomes in acute ischemic stroke patients.

## 1. Introduction

Stroke is the second leading cause of death and a major cause of disability worldwide [1]. The global incidence of stroke has increased over the last few decades, with the most common subtype being ischemic stroke [2]. Patients require considerable time to recover from stroke and may continue to experience disability even after adequate treatment. Studies have therefore assessed the ability of various biological markers to predict stroke prognosis. Fibrinogen levels have been associated with early neurological deterioration in patients with both acute ischemic stroke and diabetes mellitus [3]. Interleukin-6 (IL-6) concentrations have been shown to correlate with stroke severity, functional outcomes within 1 year, and infarct size [4].

In addition, inflammatory markers have been reported to be predictors of stroke. Inflammation after ischemic stroke plays an essential role in the pathogenesis of brain damage. Various types of inflammatory cells, including neutrophils, lymphocytes, and monocytes, are recruited to ischemic brain tissues, where they produce several types of inflammatory mediators [5]. Inflammatory biomarkers have been associated with stroke severity and clinical outcomes. For example, elevated leukocyte and neutrophil counts have been associated with larger infarct volume [6], and elevated leukocyte counts have been associated with greater initial stroke severity and poorer short- and long-term clinical outcomes [7,8]. Moreover, increased serum concentrations of high-sensitivity C-reactive protein (hsCRP) were found to correlate with an increased risk of stroke recurrence and poorer functional outcomes within 3 months [9].

Neutrophil-to-lymphocyte ratio (NLR) is an emerging biomarker for assessing the systemic inflammatory status of an individual, including in patients with cardiovascular disease, peripheral vascular disease, and cancer. NLR has also been associated with the prognosis of patients with acute ischemic stroke. For example, a higher NLR has been associated with greater initial stroke severity and a higher short-term mortality rate [10,11], as well as with poorer short-term functional outcomes and an increased risk of recurrent ischemic stroke [12,13]. A higher NLR has also been associated with several post-stroke complications, including higher risks of symptomatic hemorrhagic transformation and symptomatic intracerebral hemorrhage [14,15]. 

Although studies to date have demonstrated that NLR is related to the severity of acute ischemic stroke and short-term functional outcomes, previous evaluations of functional outcomes have been limited. For example, indicators of physical performance, such as upper and lower limb function, have not been evaluated, and few studies have evaluated the relationship between NLR and the cognitive function of acute ischemic stroke patients. The present study therefore assessed the relationship of NLR with physical and functional performance, as determined using the Berg Balance Scale (BBS) and the Manual Function Test (MFT), in acute ischemic stroke patients. In addition, the relationship of NLR with cognitive function in these patients was analyzed using the Korean Mini-Mental State Examination (K-MMSE). The overall functional status was assessed using the Korean version of the modified Barthel Index (K-MBI) and the modified Rankin Scale (mRS). The present study was therefore designed to comprehensively investigate the associations of NLR with short-term functional outcomes in patients with acute ischemic stroke.

## 2. Materials and Methods

### 2.1. Study Population and Design

This retrospective study included all patients who had been diagnosed with acute ischemic stroke within 7 days of onset and admitted to the Kyung Hee University Medical Center, Seoul, South Korea, from March 2020 to May 2022. The medical records of 1427 patients admitted to this hospital for acute ischemic stroke during this time period were reviewed. Of these patients, 328 were excluded due to a history of cerebrovascular diseases, hematologic diseases, malignant tumors, inflammatory or infectious diseases, immunosuppressant use, severe renal or hepatic diseases, neurocognitive disorders, or impaired physical conditions (Figure 1). Patients who had been initially treated at other hospitals and then transferred to Kyung Hee University Hospital were also excluded. The baseline demographic and clinical characteristics of the patients were obtained from their medical records. The factors recorded at admission included demographic characteristics, such as age, gender, and body mass index; medical history, such as hypertension, diabetes mellitus, dyslipidemia, and coronary artery disease; lifestyle factors, including smoking status and alcohol intake; and laboratory data (Table 1). Blood tests were performed within 24 h after admission for all subjects, with complete blood counts (CBC) performed to assess white blood cell, neutrophil, and lymphocyte counts. NLR was calculated as the ratio of neutrophil-to-lymphocyte counts. Other laboratory data included hemoglobin concentration; platelet and monocyte counts; and serum concentrations of hsCRP, total cholesterol, triglycerides, high-density lipoproteins, and low-density lipoproteins. The performance of reperfusion therapy (intravenous thrombolysis or endovascular therapy) during admission was also recorded (Table 1).

### 2.2. Outcome Measurements

Stroke severity was measured on admission using the National Institutes of Health Stroke Scale (NIHSS). The NIHSS is a stroke severity measurement scale consisting of 11 items that evaluate the level of consciousness, gaze, visual fields, facial palsy, motor arm, motor leg, ataxia, sensory, language, dysarthria, extinction, and inattention [16]. Patients were categorized into two groups based on their NIHSS scores at admission: moderate-to-severe stroke (≥5 points) and mild stroke (<5 points) [17]. All patients underwent physical function and cognitive tests within 2 weeks of stroke onset. Cognitive function was evaluated using the K-MMSE, a Korean version of the MMSE that evaluates seven areas: time orientation, spatial orientation, memory registration, attention and calculation, memory recall, language, and space-time configuration. Total scores on the K-MMSE range from 0 to 30, with scores ≤ 23 defined as unfavorable outcomes because they indicate cognitive impairment [18]. Dynamic balance was evaluated using the BBS, a valid and reliable instrument for measuring both the static and dynamic aspects of balance in both the elderly and in stroke patients. BBS scores range from 0 to 56 points, with higher scores reflecting better balance. BBS scores ≤ 40 are indicative of a higher fall risk and are considered unfavorable outcomes [19]. ADL performance was evaluated using the K-MBI, a Korean version of the MBI that evaluates an individual’s degree of independence. The MBI covers 10 domains of functioning (activities): bowel control; bladder control; need for help with grooming, toilet use, feeding, transfers, walking, dressing, climbing stairs, and bathing [20]. K-MBI scores ≤ 60 indicate severe to total dependence on assistance, and were defined as unfavorable outcomes [21]. The mRS score, which reflects general disability and stroke patients’ need for assistance, was also evaluated at discharge. An mRS score ≥ 3 was defined as an unfavorable outcome, as it represents moderate-to-severe disability [22]. The MFT of the hemiplegic hand was also evaluated within 2 weeks of stroke onset. The MFT, which evaluates the motor function of the upper extremities in stroke patients, consists of eight items, including forward elevation of the arm, lateral elevation of the arm, touching the occiput with the palm, touching the back with the palm, grasping, pinching, carrying cubes, and pegboard manipulations. Total scores range from 0 to 32 points [23], with MFT scores ≤ 13 indicating unfavorable outcomes [24]. The BBS, MFT, K-MMSE, K-MBI, NIHSS, and mRS were evaluated in patients by experienced therapists and clinicians. The study protocol was approved by the Institutional Review Board of Kyung Hee University Hospital (KHUH 2022-09-024), which waived the requirement for written informed consent due to the retrospective nature of this study.

### 2.3. Statistical Analysis

The Kolmogorov–Smirnov test was used to evaluate the normal distribution of continuous variables. Normally distributed variables were compared by analysis of variance (ANOVA), and non-normally distributed variables by Kruskal–Wallis tests. Categorical variables were compared by chi-square tests. Correlations between two continuous variables were analyzed by Spearman’s correlation tests. Univariate and multivariate logistic analyses were performed to evaluate the associations between NLR and clinical outcomes. The results of the multivariate analyses were adjusted for potential confounding factors, including age, sex, body mass index, hypertension, diabetes mellitus, dyslipidemia, coronary artery disease, smoking, alcohol intake, intravenous thrombolysis, endovascular therapy, time from stroke onset to hospital admission, hemoglobin concentration, white blood cell count, platelet count, monocyte count, serum concentrations of hsCRP, total cholesterol, triglycerides, high-density lipoproteins, and low-density lipoproteins. Acute stroke patients were divided into NLR tertiles (<1.84, 1.84–2.71, and >2.71), with mean stroke severity and outcomes in the three tertile groups compared by independent sample *t*-tests. The optimal NLR cut-off values for stroke severity and functional outcomes were determined by a receiver operating characteristic (ROC) curve analysis. All statistical analyses were performed using SPSS Statistics 25 (SPSS Inc., Chicago, IL, USA), with two-sided *p*-values < 0.05 defined as statistically significant. 

## 3. Results

Of the 1427 patients with acute ischemic stroke admitted to KHUH during the study period, 414 were excluded, including 328 with a history of cerebrovascular diseases, hematologic diseases, malignant tumors, inflammatory or infectious diseases, immunosuppressant use, severe renal or hepatic diseases, neurocognitive disorders, or impaired physical conditions, and 46 who were initially treated at other hospitals. In addition, 812 patients with missing follow-up data were excluded. The present study included 201 patients (Figure 1). Based on their NLR at admission, patients were grouped into three tertiles: the bottom (NLR < 1.84, n = 67), middle (NLR 1.84–2.71, n = 67), and top (NLR > 2.71, n = 67) tertiles. The baseline characteristics of the total patient population and of the three tertiles are summarized in Table 1. The median patient age was 67 years (range, 58–75 years), and 68 (33.8%) of the 201 patients were women. The white blood cell and neutrophil counts were significantly higher, and lymphocyte counts were significantly lower, in the top tertile group. Other patient characteristics, including demographic factors, medical history, laboratory data, time from stroke onset to hospital admission time, and reperfusion therapy, did not differ significantly among the three tertile groups. 

Table 2 shows the odds ratios for stroke severity and functional outcomes in the three NLR tertile groups. The univariable analysis showed that, compared with the lowest NLR tertile group, patients in the highest NLR tertile group had higher risks of unfavorable outcomes on the K-MBI (OR, 3.21; 95% CI, 1.59–6.52; *p* = 0.001), BBS (OR, 2.45; 95% CI, 1.20–5.01; *p* = 0.014), and K-MMSE (OR, 2.40; 95% CI, 1.07–5.35; *p* = 0.033). In the multivariate analysis, after adjusting for other confounding factors, patients in the top tertile group had higher risks of unfavorable outcomes on the K-MBI (OR, 3.47; 95% CI, 1.35–8.87; *p* = 0.010) and K-MMSE (OR, 3.26; 95% CI, 1.13–9.45; *p* = 0.029). The NIHSS scores tended to be higher in the top than in the bottom tertile group, but this difference was not statistically significant. 

The results of comparing stroke severity and functional outcomes among tertile groups using independent sample *t*-tests are summarized in Table 3. Compared with the bottom tertile group, the patients in the top tertile group had significantly lower mean BBS (mean difference = 8.19, *p* = 0.009), MFT (mean difference = 3.15, *p* = 0.041), and K-MBI (mean difference = 15.05, *p* = 0.009) scores. Moreover, compared with the middle tertile group, the patients in the top tertile group had significantly lower K-MBI (mean difference = 8.85, *p* = 0.042) and higher mRS (mean difference = 0.39, *p* = 0.037) scores. The analysis of continuous variables using the Spearman correlation test revealed a significant positive correlation between NLR and NIHSS score (ρ = 0.176, *p* = 0.013) and negative correlations of NLR with MFT of the hemiplegic hand (ρ = −0.237, *p* = 0.001), BBS (ρ = −0.203, *p* = 0.004), K-MBI (ρ = −0.243, *p* = 0.001), and K-MMSE (ρ = −0.182, *p* = 0.010) scores (Table 4).

The optimal cut-offs of NLR for stroke severity and functional outcomes were determined by a receiver operating characteristic (ROC) curve analysis (Figure 2). The NLR cut-offs were 2.09 for moderate-to-severe stroke severity, 2.05 each for unfavorable BBS and K-MBI scores, and 2.01 for unfavorable K-MMSE scores. The multivariable logistic regression analyses of the relationships between NLR cut-offs for stroke severity and functional outcomes showed that higher NLR levels were significantly associated with increased stroke severity, as measured by the NIHSS (OR, 3.03; 95% CI, 1.29–7.14; *p* = 0.011) and unfavorable outcomes on the K-MBI (OR, 3.31; 95% CI, 1.55–7.07; *p* = 0.002) and K-MMSE (OR, 4.80; 95% CI, 1.90–12.14; *p* = 0.001) (Table 5).

## 4. Discussion

Although an association between the NLR and the prognosis of acute ischemic stroke patients has been demonstrated in previous studies, less is known about the associations of NLR with functional outcomes in these patients. The present study analyzed the relationship between NLR and short-term functional outcomes, such as MFT, BBS, K-MMSE, K-MBI, and mRS scores, in acute ischemic stroke patients. A multivariate logistic regression analysis based on NLR tertiles showed that a higher NLR was associated with poorer outcomes according to K-MBI and K-MMSE scores, indicating that a higher NLR was associated with an overall lower ADL and cognitive performance, respectively. NLR was positively correlated with initial stroke severity, as measured by the NIHSS, and was negatively correlated with post-stroke functional outcomes, including the MFT of the hemiplegic hand and BBS, K-MBI, and K-MMSE scores. Multivariate regression analyses based on the optimal NLR cut-off values determined by an ROC curve analysis also showed that higher NLR levels were associated with increased NIHSS scores and poorer functional outcomes according to K-MBI and K-MMSE scores. These results indicate that a higher NLR is associated with poorer short-term physical and cognitive performance in patients with acute ischemic stroke. Sometimes, health-related evaluation tools that have been verified and commonly used in foreign countries cannot be directly applied to another country, especially with a different language setting. In order to use the tools appropriately, modifications according to the country’s cultural backgrounds and the language setting must be considered [25]. The K-MMSE and K-MBI have been widely used in previous studies as tools for screening patients with cognitive impairment and assessing the levels of daily living ability of stroke patients in Korea, with high reliabilities and validities [26,27]. As both tests have the similar internal consistency reliabilities with the international versions, the potential impact arising from the difference between the international version and the Korean version of the tests may be somewhat clinically insignificant to the results of our study [26,27,28]. 

Inflammation after stroke is associated with the pathogenesis of ischemic brain injury. Inflammatory status can affect the clinical outcomes and the prognosis of stroke patients [29]. Although studies have evaluated the mechanisms underlying inflammatory and immunological responses after acute ischemic stroke, the precise pathogenic pathways remain incompletely understood. The activation of various types of inflammatory cells, including microglia, neutrophils, lymphocytes, and macrophages, is important for the pathogenesis of the inflammatory response after ischemic brain injury [5]. Inflammatory mediators released from ischemic brain tissues promote the expression of adhesion molecules on endothelial cells, accompanied by the recruitment of leukocytes to damaged brain lesions [30,31]. During the acute phase of ischemic brain injury, microglial cells produce various cytokines, which can potentiate the further infiltration of leukocytes [32,33]. Neutrophils, the first type of leukocytes to infiltrate damaged brain tissue during the subacute phase of ischemic brain injury [34], produce toxic amounts of reactive oxygen species, along with inflammatory mediators, further accelerating the inflammatory process. Among these inflammatory mediators is matrix metalloproteinase-9 (MMP-9), which plays important roles in disrupting the blood–brain barrier and in hemorrhagic transformation [35]. Several types of lymphocytes are involved in the later phases of the inflammatory process after acute ischemic stroke. In particular, regulatory T cells exert neuroprotective effects on damaged ischemic brain lesions by producing anti-inflammatory cytokines and suppressing further inflammatory processes [36]. Interleukin-10 (IL-10), which is produced by regulatory T cells, inhibits post-ischemic inflammatory pathways by downregulating pro-inflammatory cytokines [37]. 

NLR is a systemic inflammatory biomarker that reflects the balance between circulating neutrophils and lymphocytes. NLR at hospital admission has been shown to correlate positively with the NIHSS scores of acute ischemic stroke patients, with a higher NLR being associated with an increased risk of 60-day mortality [11]. An increased NLR was also shown to be associated with unfavorable outcomes on the modified Rankin Scale 3 months after stroke onset, as well as with unfavorable outcomes on the modified Barthel Index at discharge [12,13,38]. Cognitive impairment 3 months after stroke onset was found to correlate with higher NLR levels [39]. A recent systematic review and meta-analysis revealed that early neurological deterioration in stroke patients was associated with a higher NLR [40]. To our knowledge, the present study is the first to demonstrate the relationship between NLR and physical performance in patients with acute ischemic stroke by incorporating the Berg Balance Scale (BBS) and the Manual Function Test (MFT). Several studies have shown a relationship between physical function and inflammatory biomarkers in older people. Higher levels of C-reactive protein (CRP) and IL-6 in people aged over 65 years were associated with poor physical performance, as determined by walking speed, chair-stand tests, and standing balance tests, along with decreased hand-grip strength (HGS) [41]. Elevated CRP levels were associated with poorer chair-stand performance and HGS loss in older people during hospitalization [42], suggesting that higher levels of inflammation are associated with decreased physical function and hand power. This finding is consistent with the present results showing associations between NLR and BBS and MFT scores. A higher degree of inflammation in adults has been associated with lower muscle strength and mass [43], which may impair overall balance and hand function [24,44]. These findings may help to explain the negative correlations of NLR with BBS and MFT scores observed in the present study. Other clinical studies have also demonstrated a relationship between inflammation and cognitive function. For example, in the Northern Manhattan Study, higher IL-6 levels were associated with poorer cognitive function, as assessed by the MMSE [45], and interleukin-12 (IL-12) levels were found to be elevated in ischemic stroke patients with cognitive decline [46]. Taken together, these results suggest that the degree of inflammation may be associated with physical and cognitive function.

The results of the present study may be explained by the underlying inflammatory processes that occur after acute ischemic stroke. Neuroinflammation after stroke can have a detrimental effect on the early-stage progression of ischemic brain injury. This neuroinflammation is mediated by the disruption of the blood–brain barrier, which leads to brain edema and secondary ischemic brain damage [47]. NLR, which reflects the combined activities of neutrophils and lymphocytes, is an indicator of the inflammatory status of ischemic stroke patients. A higher NLR is associated with a more severe degree of inflammation, which may result in poorer functional outcomes. 

Similar to the present study, several other recent studies have shown that NLR is associated with outcomes in acute ischemic stroke patients. For example, an NLR cut-off of 6.4 was found to be significantly associated with an increased risk of early neurological deterioration after endovascular treatment (OR, 1.011; 95% CI, 1.04–1.18; *p* = 0.001) [48], and an NLR cut-off of 6.2 was found to be associated with unsuccessful reperfusion after endovascular treatment (OR, 1.11; 95% CI, 1.04–1.19; *p* = 0.001) [49]. The combination of a high NLR with a low lymphocyte-to-monocyte ratio (LMR) was found to be related to poor functional outcomes on the mRS at 3 months (OR, 3.407; 95% CI, 1.449–8.011; *p* = 0.005), with NLR and LMR cut-offs of 5.73 and 2.08, respectively [50]. Regarding carotid plaque, a major cause of acute ischemic stroke, a retrospective study revealed that NLR cut-offs of 3.47 and 4.41 were associated with increased risks of 12-month restenosis (OR, 34.22; 95% CI, 14.99–78.12; *p* < 0.001) and mortality (OR, 37.62; 95% CI, 12.87–109.97; *p* < 0.001), respectively, after carotid endarterectomy [51]. Although the cut-off values of NLR in these studies differed from the cut-off values in the present study, the results were similar in that higher values of NLR were related to poor functional outcomes. Furthermore, higher values of NLR were found to be associated with increased risks of acute deep vein thrombosis after total knee arthroplasty [52] and in-hospital mortality after ST-elevation myocardial infarction [53]. Taken together, these findings indicate that NLR values may be associated with the outcomes of thromboembolic events other than acute ischemic stroke, suggesting that NLR may be an effective and useful biomarker. 

The present study had several limitations, including its retrospective design and the inclusion of patients from a single hospital, which may have resulted in a selection bias, such as a sampling bias. In addition, only about 15% of the patients with acute ischemic stroke admitted to KHUH fulfilled the inclusion and exclusion criteria. Furthermore, NLR was measured only at the time of hospital admission; dynamic changes in NLR reflecting the time course of neuroinflammation were not evaluated. Longitudinal measurements of NLR from hospitalization to discharge may help to better understand the relationship between inflammation and ischemic stroke. Finally, the present study did not evaluate lesion locations or size. The stroke subtype could influence the prognosis of ischemic stroke patients, acting as a potential confounding factor for the relationship between NLR and post-stroke functional outcomes. 

## 5. Conclusions

NLR is a simple and cost-effective biomarker that may reflect underlying inflammation after stroke. The results of the present study demonstrated that a higher NLR was associated with poorer short-term functional outcomes, as determined by K-MBI and K-MMSE scores, in patients with acute ischemic stroke. NLR may be a useful predictor of short-term functional outcomes in patients with acute ischemic stroke.

## Figures and Tables

**Figure 1 ijerph-20-00898-f001:**
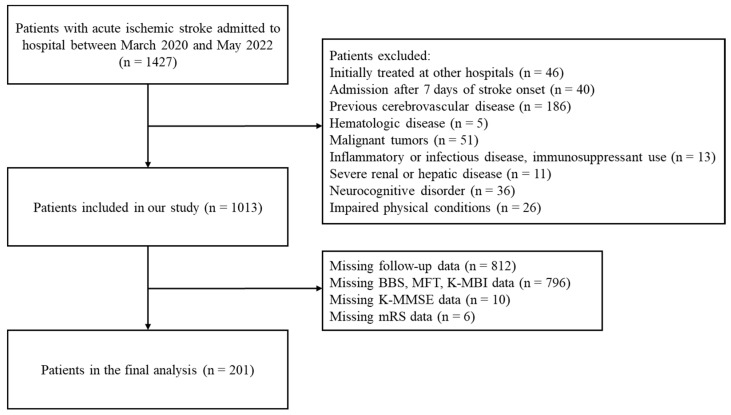
Flow diagram for patient selection.

**Figure 2 ijerph-20-00898-f002:**
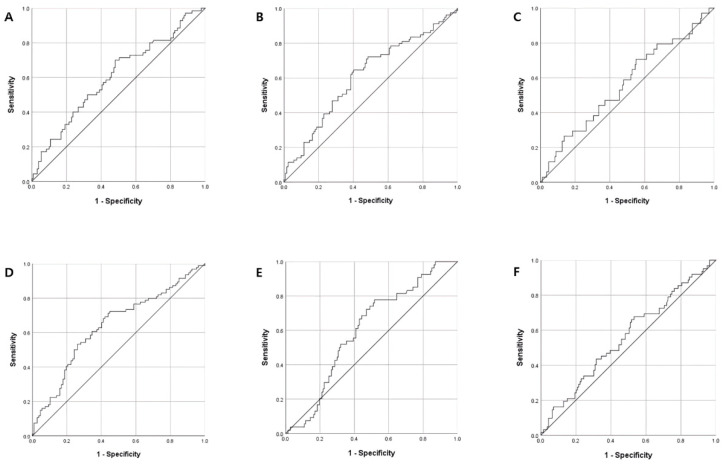
Receiver operating characteristic curve analyses of the associations of NLR with stroke severity and functional outcomes. (**A**) NIHSS (AUC: 0.61, *p* = 0.014), (**B**) BBS (AUC: 0.61, *p* = 0.007), (**C**) MFT (AUC: 0.56, *p* = 0.255), (**D**) K-MBI (AUC: 0.64, *p* = 0.001), (**E**) K-MMSE (AUC: 0.60, *p* = 0.025), (**F**) mRS (AUC: 0.60, *p* = 0.181). Abbreviations: NIHSS, National Institutes of Health Stroke Scale; BBS, Berg Balance Scale; MFT, Manual Function Test; K-MBI, Korean version of the modified Barthel Index; K-MMSE, Korean Mini-Mental State Examination; mRS, modified Rankin Scale; AUC, area under the curve.

**Table 1 ijerph-20-00898-t001:** Baseline characteristics according to tertiles of NLR.

	Total	Tertile 1 (NLR < 1.84)	Tertile 2 (NLR 1.84–2.71)	Tertile 3 (NLR > 2.71)	
	(n = 201)	(n = 67)	(n = 67)	(n = 67)	*p*-Value
Demographic characteristics					
Age (years)	67 (58–75)	66 (59–75)	66 (58–74)	69 (56–77)	0.852
Gender (female)	68 (33.8)	24 (35.8)	26 (38.8)	18 (26.9)	0.141
BMI (kg/m^2^)	24.4 (22.4–27.1)	24.6 (22.3–28.0)	24.0 (22.5–26.2)	24.4 (22.1–26.8)	0.585
Medical history					
Hypertension (n)	111 (55.2)	33 (49.3)	37 (55.2)	41 (61.2)	0.165
Diabetes mellitus (n)	50 (24.9)	15 (22.4)	19 (28.4)	16 (23.9)	0.427
Dyslipidemia (n)	45 (22.4)	15 (22.4)	19 (28.4)	11 (16.4)	0.097
Coronary artery disease (n)	13 (6.5)	5 (7.5)	3 (4.5)	5 (7.5)	0.466
Smoking (n)	86 (42.8)	28 (41.8)	29 (43.3)	29 (43.3)	0.861
Alcohol intake (n)	90 (44.8)	26 (38.8)	31 (46.3)	33 (49.3)	0.224
Intravenous thrombolysis (n)	17 (8.5)	8 (11.9)	7 (10.4)	2 (3)	0.067
Endovascular therapy (n)	16 (8)	7 (10.4)	5 (7.5)	4 (6)	0.351
Time from stroke onset to hospital admission (days)	2 (1–2)	2 (1–2)	2 (1–2)	2 (1–2)	0.970
Laboratory findings					
Hemoglobin (g/dL)	14.4 ± 1.7	14.4 ± 1.5	14.4 ± 1.8	14.4 ± 1.7	0.983
Platelet count (10^9^/L)	225 (191–269)	221 (189–249)	225 (202–272)	237 (190–292)	0.258
WBC count (10^9^/L)	7.3 (6.2–9.2)	6.7 (5.3–8.1)	7.0 (6.3–8.6)	8.6 (7.1–10.7)	<0.001 ***
Neutrophil count (10^9^/L)	4.5 (3.5–6.1)	3.4 (2.7–4.3)	4.4 (3.8–5.4)	6.2 (5.0–8.2)	<0.001 ***
Lymphocyte count (10^9^/L)	2.0 (1.5–2.6)	2.6 (2.0–3.0)	2.0 (1.7–2.5)	1.5 (1.2–1.9)	<0.001 ***
Monocyte count (10^9^/L)	0.4 (0.3–0.5)	0.4 (0.3–0.5)	0.4 (0.3–0.4)	0.4 (0.3–0.5)	0.804
hsCRP (mg/L)	0.1 (0.1–0.3)	0.1 (0.1–0.3)	0.1 (0.1–0.3)	0.2 (0.1–0.4)	0.223
HbA1c (%)	5.9 (5.5–6.5)	5.9 (5.5–6.4)	5.9 (5.5–6.5)	5.9 (5.5–6.6)	0.959
TC (mg/dL)	202 ± 49	195 ± 48	202 ± 46	208 ± 52	0.315
TG (mg/dL)	140 (102–205)	164 (111–232)	126 (99–172)	145 (100–189)	0.055
HDL (mg/dL)	49 (42–60)	48 (42–57)	50 (41–62)	51 (42–60)	0.525
LDL (mg/dL)	125 ± 37	117 ± 33	124 ± 39	132 ± 37	0.051

Normally distributed continuous variables are presented as mean ± standard deviation (SD); non-normally distributed continuous variables as median (interquartile range [IQR]); and categorical variables as numbers and percentages. Abbreviations: BMI, body mass index; WBC, white blood cell; hsCRP, high-sensitivity C-reactive protein; HbA1c, hemoglobin A1c; TC, total cholesterol; TG, triglycerides; HDL, high-density lipoprotein; LDL, low-density lipoprotein. *** *p* < 0.001.

**Table 2 ijerph-20-00898-t002:** Risks of stroke severity and functional outcomes according to NLR tertile.

	Tertile 1 (NLR < 1.84)	Tertile 2 (NLR 1.84–2.71)	Tertile 3 (NLR > 2.71)
	(n = 67)	(n = 67)	(n = 67)
Moderate-to-severe stroke severity (NIHSS ≥ 5)			
Crude OR (95% CI)	ref	1.15 (0.55–2.42, *p* = 0.706)	2.05 (1.00–4.20, *p* = 0.050)
Adjusted OR (95% CI)	ref	1.11 (0.44–2.81, *p* = 0.827)	2.52 (0.91–6.97, *p* = 0.075)
Unfavorable BBS outcome (BBS ≤ 40)			
Crude OR (95% CI)	ref	1.71 (0.83–3.51, *p* = 0.147)	2.45 (1.20–5.01, *p* = 0.014) *
Adjusted OR (95% CI)	ref	1.32 (0.57–3.05, *p* = 0.516)	1.55 (0.60–3.97, *p* = 0.365)
Unfavorable MFT outcome (MFT ≤ 13)			
Crude OR (95% CI)	ref	1.41 (0.55–3.60, *p* = 0.477)	1.55 (0.61–3.92, *p* = 0.353)
Adjusted OR (95% CI)	ref	0.96 (0.32–2.88, *p* = 0.940)	0.96 (0.29–3.20, *p* = 0.940)
Unfavorable K-MBI outcome (MBI ≤ 60)			
Crude OR (95% CI)	ref	1.46 (0.73–2.94, *p* = 0.288)	3.21 (1.59–6.52, *p* = 0.001) **
Adjusted OR (95% CI)	ref	1.38 (0.60–3.13, *p* = 0.448)	3.47 (1.35–8.87, *p* = 0.010) *
Unfavorable K-MMSE outcome (MMSE ≤ 23)			
Crude OR (95% CI)	ref	1.81 (0.80–4.12, *p* = 0.154)	2.40 (1.07–5.35, *p* = 0.033) *
Adjusted OR (95% CI)	ref	2.38 (0.91–6.22, *p* = 0.077)	3.26 (1.13–9.45, *p* = 0.029) *
Unfavorable mRS outcome (mRS ≥ 3)			
Crude OR (95% CI)	ref	1.00 (0.47–2.12, *p* = 1.000)	1.41 (0.68–2.92, *p* = 0.356)
Adjusted OR (95% CI)	ref	0.74 (0.30–1.84, *p* = 0.521)	0.83 (0.31–2.22, *p* = 0.704)

Abbreviations: NIHSS, National Institutes of Health Stroke Scale; BBS, Berg Balance Scale; MFT, Manual Function Test; K-MBI, Korean version of the modified Barthel Index; K-MMSE, Korean Mini-Mental State Examination; mRS, modified Rankin Scale; OR, odds ratio; CI, confidence interval. Adjusted for age, sex, BMI, hypertension, diabetes, dyslipidemia, coronary artery disease, smoking, alcohol intake, intravenous thrombolysis, endovascular therapy, onset-to-admission time, hemoglobin, WBC, platelet, monocyte count, hsCRP, TC, TG, HDL, and LDL. * *p* < 0.05, ** *p* < 0.01.

**Table 3 ijerph-20-00898-t003:** Comparison of stroke severity and functional outcomes in patients categorized by NLR tertiles.

	Tertile 1 (NLR < 1.84)	Tertile 2 (NLR 1.84–2.71)	Tertile 3 (NLR > 2.71)			
	(n = 67)	(n = 67)	(n = 67)	P1	P2	P3
NIHSS	3.73 ± 3.47	3.75 ± 3.71	4.46 ± 3.06	0.981	0.198	0.225
BBS	43.16 ± 15.64	38.49 ± 19.28	35.27 ± 18.67	0.126	0.009 **	0.327
MFT	24.03 ± 8.50	22.64 ± 9.27	20.88 ± 9.12	0.368	0.041 *	0.270
K-MBI	68.72 ± 24.76	62.61 ± 25.82	53.67 ± 24.48	0.165	0.001 **	0.042 *
K-MMSE	25.33 ± 6.20	24.97 ± 4.95	23.00 ± 7.35	0.712	0.050	0.072
mRS	1.91 ± 1.04	1.85 ± 1.09	2.24 ± 1.05	0.746	0.071	0.037 *

Results are presented as mean ± standard deviation (SD). P1, comparison between tertiles 1 and 2; P2, comparison between tertiles 1 and 3; P3, comparison between tertiles 2 and 3; Abbreviations: NIHSS, National Institutes of Health Stroke Scale; BBS, Berg Balance Scale; MFT, Manual Function Test; K-MBI, Korean version of the modified Barthel Index; K-MMSE, Korean Mini-Mental State Examination; mRS, modified Rankin Scale. * *p* <0.05, ** *p* < 0.01.

**Table 4 ijerph-20-00898-t004:** Correlations of NLR with stroke severity and functional outcomes.

Variables	Spearman Correlation Coefficient (ρ)	*p*-Value
NIHSS	0.176	0.013 *
BBS	−0.203	0.004 **
MFT	−0.237	0.001 **
K-MBI	−0.243	0.001 **
K-MMSE	−0.182	0.010 *
mRS	0.121	0.086

Abbreviations: NIHSS, National Institutes of Health Stroke Scale; BBS, Berg Balance Scale; MFT, Manual Function Test; K-MBI, Korean version of the modified Barthel Index; K-MMSE, Korean Mini-Mental State Examination; mRS, modified Rankin Scale. * *p* < 0.05, ** *p* < 0.01.

**Table 5 ijerph-20-00898-t005:** Logistic regression analysis of the associations between the optimal NLR cut-offs and the indicators of stroke severity and functional outcomes.

Variables	AUC (95% CI)	Cut-Off Value	Sensitivity	Specificity	Crude OR (95% CI)	Adjusted OR (95% CI)
NIHSS	0.61 (0.52–0.69, *p* = 0.014) *	2.09	0.70	0.52	2.44 (1.32–4.52, *p* = 0.004) **	3.03 (1.29–7.14, *p* = 0.011) *
BBS	0.61 (0.53–0.69, *p* = 0.007) *	2.05	0.72	0.52	2.77 (1.51–5.08, *p* = 0.001) **	2.07 (0.96–4.49, *p* = 0.065)
MFT	0.56 (0.45–0.67, *p* = 0.255)	2.64	0.44	0.67	1.52 (0.72–3.22, *p* = 0.270)	1.23 (0.46–3.33, *p* = 0.684)
K-MBI	0.64 (0.56–0.72, *p* = 0.001) *	2.05	0.72	0.55	3.22 (1.78–5.81, *p* < 0.001) ***	3.31 (1.55–7.07, *p* = 0.002) **
K-MMSE	0.60 (0.52–0.69, *p* = 0.025) *	2.01	0.78	0.48	3.18 (1.55–6.53, *p* = 0.002) **	4.80 (1.90–12.14, *p* = 0.001) **
mRS	0.60 (0.47–0.65, *p* = 0.181)	2.05	0.68	0.47	1.85 (0.98–3.46, *p* = 0.056)	1.19 (0.53–2.67, *p* = 0.668)

Abbreviations: NIHSS, National Institutes of Health Stroke Scale; BBS, Berg Balance Scale; MFT, Manual Function Test; K-MBI, Korean version of the modified Barthel Index; K-MMSE, Korean Mini-Mental State Examination; mRS, modified Rankin Scale; AUC, area under the curve; OR, odds ratio; CI, confidence interval. Results adjusted for age, sex, BMI, hypertension, diabetes, dyslipidemia, coronary artery disease, smoking, alcohol intake, intravenous thrombolysis, endovascular therapy, onset-to-admission time, hemoglobin; WBC, platelet, and monocyte counts; and hsCRP, TC, TG, HDL, and LDL concentrations. * *p* < 0.05, ** *p* < 0.01, *** *p* < 0.001.

## Data Availability

Not applicable.

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
