# Peer review of "Neutrophil-to-Lymphocyte Ratio as a Predictor of Short-Term Functional Outcomes in Acute Ischemic Stroke Patients"

_ijerph, 2023, doi:10.3390/ijerph20020898_

Round 1

Reviewer 1 Report

I would like to congratulate the authors on their work! This is potentially significant research since it provides evidence regarding the association between NLR and short-term functional outcomes in patients with acute ischemic stroke in a more detailed and comprehensive manner, by measuring several indicators.

However, there are some aspects that must be addressed.

1. Please read the Author's Instructions and format the Abstract as follows: Background, Methods, Results, and Conclusions.

2. English needs to be slightly improved and typo errors corrected.

Line 110: items that evaluates level -> items that evaluates the level

Line 127: bladder control, requirement for help -> bladder control, the requirement for help

Line 134: to evaluate motor function -> to evaluate the motor function

And so on... all over the manuscript.

3. Please put all table titles in line with the layout of the tables. This is available for abbreviations under the tables also.

4. I suggest authors to make the ROC analyze regarding the NLR, to identify the optimal cut-off value and after that to analyze the multivariate regression to see if the high baseline NLR value is an independent predictor for short-term functional outcomes in AIS.

4. Because biomarkers are a very interesting topic that has received a lot of attention in the latest years, I strongly advise the authors to improve the quality of the research by comparing the results with articles that are new (from 2022), regarding the thromboembolic events to see if it is a difference, in the Discussion section. See the following list as examples:

- https://doi.org/10.3390/ijerph192113934

- https://doi.org/10.3390/life12091415

- https://doi.org/10.3390/medicina58101502

- https://doi.org/10.3390/jpm12081221

- https://doi.org/10.3390/jpm11080696

5. Please avoid being repetitive with the formulations in the Discussion section (see “Prior studies”, “Our study”) and keep this section based on comparing the values of this research with the others in the literature. Don’t repeat information from the Introduction and, compare the actual values (numbers) from the other articles with the values that have resulted in this study.

Reviewer 2 Report

The authors tried to investigate the prognosis relationship of NLR with stroke severity outcomes including into analysis more detailed functional and cognitive scored data. They found that higher NLR is associated with poor short term physical and cognitive performance.  When adjusting for other potential confounding factors they included too many factors considering the overall patient number. However, the authors mentioned it as a limitation of the study.  

The Tertile 3, thus a NLR > 2.71 has a statistical power to predict poor oucomes and  should be concidered as a high ratio. In the Tertile 2 the predictive power is poor.

Reviewer 3 Report

Dear authors,

I have read with interest your article, and I find your study to be fair and well-conducted. Still, I have an important issue to address: you say that NLR is an emerging novel biomarker, yet in your paper, there is no discussion regarding the newly published articles, with only 4 references from 2021, and none from 2022. There are many papers published in this field of work in 2022, and I have found quickly even a Systematic review and meta-analysis on this topic, published by Sarejloo S et al (I have no ties with this author, it is not mandatory to discuss their work).

There are many other papers on the topic of NLR-stroke. Of course, you have evaluated many aspects by different questionnaires, and this is a strong point of your work - you should acknowledge also the strengths of your study, not only the limitations.

I feel there is still room for improvement in the Discussion section with many other papers that could be cited. 

Good luck!

Round 2

Reviewer 1 Report

Excellent work. I want to congratulate the authors.

Author Response

We would like to thank the editor and reviewers for their helpful comments and suggestions.